

# Shades of yellow: interactive effects of visual and odour cues in a pest beetle

Sarah E.J. Arnold[1], Philip C. Stevenson[1,2] and Steven R. Belmain[1]

[1] Natural Resources Institute, University of Greenwich, Chatham Maritime, Kent, United Kingdom
[2] Royal Botanic Gardens, Kew, Richmond, Surrey, United Kingdom

## ABSTRACT

**Background:** The visual ecology of pest insects is poorly studied compared to the role of odour cues in determining their behaviour. Furthermore, the combined effects of both odour and vision on insect orientation are frequently ignored, but could impact behavioural responses.

**Methods:** A locomotion compensator was used to evaluate use of different visual stimuli by a major coleopteran pest of stored grains (*Sitophilus zeamais*), with and without the presence of host odours (known to be attractive to this species), in an open-loop setup.

**Results:** Some visual stimuli—in particular, one shade of yellow, solid black and high-contrast black-against-white stimuli—elicited positive orientation behaviour from the beetles in the absence of odour stimuli. When host odours were also present, at 90° to the source of the visual stimulus, the beetles presented with yellow and vertical black-on-white grating patterns changed their walking course and typically adopted a path intermediate between the two stimuli. The beetles presented with a solid black-on-white target continued to orient more strongly towards the visual than the odour stimulus.

**Discussion:** Visual stimuli can strongly influence orientation behaviour, even in species where use of visual cues is sometimes assumed to be unimportant, while the outcomes from exposure to multimodal stimuli are unpredictable and need to be determined under differing conditions. The importance of the two modalities of stimulus (visual and olfactory) in food location is likely to depend upon relative stimulus intensity and motivational state of the insect.

## INTRODUCTION

Understanding the cues used by pest insects to locate host material is an essential element of devising sustainable control strategies to reduce impacts on food production and storage, as well as providing insights into their ecology and evolution. While research on the olfaction of pest insects is highly developed, work on the evaluation of visual preferences among pests remains lacking, despite that these preferences may play a key role in host location for many pests (*Reeves, 2011*). This knowledge gap in pests contrasts with extensive work on colour vision in some other insect groups, in particular honeybees

Corresponding author
Sarah E.J. Arnold,
s.e.j.arnold@greenwich.ac.uk

(*Backhaus, 1993*; *Backhaus, Werner & Menzel, 1987*; *Dyer, Spaethe & Prack, 2008*), bumblebees (*Chittka & Raine, 2006*; *Dyer, Spaethe & Prack, 2008*), Drosophila (*Morante & Desplan, 2008*), and some non-pest Lepidoptera (*Bernard & Remington, 1991*; *Eguchi et al., 1982*; *Kelber & Pfaff, 1997*; *Telles et al., 2014*). Furthermore, it is often hard to ascertain the relative importance of visual and olfactory stimuli in location of host material by some insects, as comparative work is not always performed and published. However, attempting to deconstruct the use of different modalities of stimulus has value in devising control strategies for pest insects as well as answering questions about the evolution of foraging behaviour.

Colour vision, defined as the ability to discriminate wavelengths of light independently of intensity, is one of several ways insects can use visual information to orient themselves (*Vorobyev & Brandt, 1997*). There are a variety of methods that can be used to investigate insects' responses to visual stimuli. Experimental setups involving tracking of the insect's movements can be considered either as open-loop, in which the insect is presented with a visual stimulus controlled by the experimenter and the insect's directionality or wing-movements are detected and recorded (*Otálora-Luna, Perret & Guerin, 2004*), versus closed-loop, in which the insect's locomotory activity is fed back to drive the visual environment with which it is presented (*Reiser & Dickinson, 2008*). Open-loop setups are simpler to operate, but closed-loop setups may provide additional information about the insect's dynamic responses and path control. This can be of value particularly in studies relating to how insects navigate, regulate their speed, estimate distance and control pitch and yaw (*Maimon, Straw & Dickinson, 2008*; *Reiser & Dickinson, 2008*).

When studying insects during visual experiments in the laboratory, larger insects can be investigated in flight arenas (*Arnold & Chittka, 2012*), Y-mazes (*Dyer et al., 2007*; *Giurfa et al., 1996*) and flight tunnels (*Srinivasan et al., 1996*; *Willis, Avondet & Zheng, 2011*). When investigating the visual ecology of flying insects, individuals are frequently tethered and optomotor responses can thus be investigated, as has been done with Drosophila (*Maimon, Straw & Dickinson, 2008*; *Yamaguchi et al., 2008*) and locusts (*Cooter, 1979*); this is amenable to investigation via both open- and closed-loop systems.

In less sophisticated setups, coloured pan traps and sticky traps can also be used to investigate colour preferences, particularly in pest insects (*Campbell & Hanula, 2007*; *Han, Zhang & Byers, 2012*; *Lunau, 2014*). This has value if the purpose is only to establish which of a range of colours elicits the highest insect capture rates, but pan traps are imprecise for investigating colour vision itself as the environment is less well controlled. A coloured trap also depends upon insects not only approaching and investigating the trap, but attempting to land and being caught by it. Electrophysiology can accurately evaluate the responses of individual insect photoreceptors, or the whole retina, to light stimuli (*Peitsch et al., 1992*; *Telles et al., 2014*), providing information about physiological capabilities to respond to visual stimuli, but does not necessarily inform about behavioural preferences or inclinations.

In this study we used a locomotion compensator (Servosphere) in an open-loop setup to evaluate responses of the maize weevil *Sitophilus zeamais* Motschulsky

(Coleoptera: Curculionidae) to both visual and olfactory stimuli. This is a relatively new method of investigating responses of insects to coloured visual stimuli (having previously only been used to assay responses of an insect to emitted light (*Beattie et al., 2011*; *Bell et al., 1983*; *Otálora-Luna & Dickens, 2011*; *Otálora-Luna, Lapointe & Dickens, 2013*) and once to an unquantified yellow stimulus of unknown spectral composition (*Van der Ent & Visser, 1991*)). The Servosphere is a 300 mm diameter ball in a motorised support, with a camera set above it. An insect placed upon the ball can run freely in any direction on the ball's surface; it is tracked by the camera, and a processor controls the rotation of the ball (driven by servomotors) to keep the insect always at the apex of the ball (*Kramer, 1976*). The motion of the ball is detected by the equipment, and processed to permit reconstruction of the insect's walking path for analysis.

The Servosphere is thus well-suited to measuring orientation behaviour of walking insects as it allows the insect to choose its direction of taxis freely and excludes the confounding factor of thigmotaxis (*Bell & Kramer, 1980*). As the insect can never reach the stimulus source, but the behaviour it displays in trying to reach or avoid the stimulus is recorded, this is considered open-loop and permits exploration of insect behaviour in an environment controlled by the experimenter. Locomotion compensators such as the Servosphere have been used for several decades, primarily to investigate use of olfactory stimuli (host odours, sex pheromones, carbon dioxide, etc.) by insects. In some cases, the insect is tethered on a freely-rotating wire, especially for insects that fly readily, whereas in other cases the insect is untethered.

The data generated by a locomotion compensator may include the speed and directionality of an insect's movement, but also metrics such as the path straightness (which would be expected to increase where a stimulus provokes a strong sensorimotor response as the insect would become more directed in its behaviour). The direction is normally measured relative to either a fixed point on the horizontal plane of the sphere or relative to the stimulus source (e.g. "upwind" direction). A typical experiment will present stimuli in succession, e.g., still air, then a clean airflow, then an airflow with added odour, then a final period of still or clean air, and the insect's behaviour at each of these stages can be observed (*Otálora-Luna, Lapointe & Dickens, 2013*). Most insects will orient in an upwind direction when faced with clean airflow, but the speed (and thus, distance moved) will increase if the insect is subsequently presented with an attractive odour. Similarly, the insect can be presented with a visual stimulus such as a light or coloured item, either alone or in the presence of an odour cue (either in the same angular location, or separately); the locomotion compensator will provide data on the direction and nature of movement shown by the insect when such a visual cue is presented (*Van der Ent & Visser, 1991*).

However, the question of whether to use an emitted light (e.g. from a light-emitting diode (LED) or a monochromator) or a non-emitting stimulus in the design of a visual assay is not always straightforward. While using the light from an LED as the visual stimulus can accurately determine the effects of a narrow band of wavelengths, stored cereal pests and many other insects are adapted to low light conditions. Consequently, their response to a bright coloured light may not be as ecologically relevant as exposing them to a non-emitting stimulus, such as coloured paper. In our experiment, we elected to

test a selection of quantified coloured papers with the insects, evaluating responses with and without the presence of host odours.

The responses of insects to multiple or, indeed, multimodal stimuli can be diverse. Some types of response to stimuli can only be observed when another stimulus is also present, as is the case for the stronger response to regressive rather than progressive patterns in *Calliphora erythrocephala*, observed only when georeceptors in the legs of the fly are stimulated (*Horn & Knapp, 1984*). Some insect responses to multiple stimuli can be simply additive; others can be antagonistic or synergistic (*Campbell & Borden, 2009*; *Giurfa, Núñez & Backhaus, 1994*). When the stimuli are presented in a way such that they appear to conflict or contradict one another, it becomes possible to make judgements about the importance of one type of stimulus over another, as has been explored in the Colorado beetle (*Otálora-Luna, Lapointe & Dickens, 2013*) and the bumblebee (*Kunze & Gumbert, 2001*), and about factors affecting whether choices are inclined towards one stimulus, intermediate, or bimodal (*Horn & Wehner, 1975*). The relative strengths of different stimuli can be important too: in *Rhagoletis pomonella*, odour cues were largely irrelevant if the visual stimulus was strong, whereas if the visual stimulus was not strongly coloured, the intensity of the odour cue became more important to the fly in locating a food source (*Aluja & Prokopy, 1993*). Somewhat similarly, hawkmoths (*Macroglossum stellatarum*) could learn an odour discrimination task if the scented targets were of a less preferred colour, but failed to learn odours if the targets were of a more preferred colour (blue), indicating that a strong, highly preferred visual stimulus interferes with responses to the odour stimulus (*Balkenius & Kelber, 2006*).

*S. zeamais* is a major pest of stored grains across sub-Saharan Africa (*Kamanula et al., 2011*). Both adults and larvae eat cereals such as wheat, maize and rice: females bore a hole in the surface of cereal grains and seal an egg within, and the larva subsequently consumes the cereal from within (*Dobie et al., 1991*); however, the beetle will also use other food material such as pasta and dried cassava when available (*Dobie et al., 1991*). As a stored product pest, most of their activity is normally expected to take place in low light conditions, but the adults are capable of flight and dispersal, so use of both visual and odour cues in host location is unsurprising. A yellow stimulus has already been found to be attractive to *S. zeamais* in a four-arm olfactometer, especially in combination with odour (*Arnold, Stevenson & Belmain, 2015*), and there is robust evidence showing the species is attracted to various cereal odours (*Arnold, Stevenson & Belmain, 2015*; *Ukeh et al., 2010*; *Ukeh et al., 2012*). However, the viewing angle in an olfactometer makes it hard to analyse effects of contrast and edges, and there is a difference between choosing to rest on an area with particular visual characteristics and actively choosing to orient towards it in a free-walking scenario. Active attraction (i.e. directed movement towards a stimulus) is key to host material location (*Hardie, 2012*) and can be better tested in the more open-ended environment of the Servosphere.

A further advantage of the locomotion compensator is the possibility to test responses when odour and colour are combined (*Otálora-Luna, Lapointe & Dickens, 2013*). Presentation of odour and colour stimuli simultaneously or successively, and from different locations relative to the insect, can help to determine and quantify preferences

for the different stimulus types. *Otálora-Luna, Lapointe & Dickens (2013)* discovered that in the neotropical weevil *Diaprepes abbreviates*, a pest and a generalist, visual cues took precedence over odour cues when this insect had to choose between apparently conflicting visual and odour cues. They found that the presence of a visual cue in the absence of odour cues actually increased activity levels in male weevils, whereas the presence of odour cues in the absence of light did not. Interestingly, presence of a green light cue appeared to override positive attraction responses to volatiles in the perpendicular direction, indicating that the odour cues are subordinate to visual cues in this species when the two appear to contradict.

In this experiment we sought to build on previous findings (*Arnold, Stevenson & Belmain, 2015*) that the maize weevil *S. zeamais* exhibits preferences for some visual stimuli more than others. Having previously established that *S. zeamais* will spend time preferentially on a tested shade of yellow paper, we were testing several hypotheses:

1. *S. zeamais* adults will orient towards visual stimuli

  a) particularly those with long-wavelength reflection and low short-wavelength reflection and
  b) particularly those with high achromatic contrast.

2. *S. zeamais* does not orient towards all stimuli broadly perceived as "yellow" to humans equally, and consequently it cannot be assumed that all "yellow" traps will be similarly effective.

3. *S. zeamais* responds to both visual and odour cues. When they are presented simultaneously but perpendicular to one another on a horizontal plane,

  a) *S. zeamais* will be influenced in its orientation direction by the presence of the visual cue when the cue is attractive, and will orient either towards the visual cue or intermediate between the visual and odour cue sources
  b) *S. zeamais* will orient more strongly towards the odour cue when the visual cue is not attractive.

Understanding the use of visual cues, when odour cues are also present, in this species, will help to refine trapping technologies for monitoring populations of *S. zeamais* in grain stores. It will also improve understanding of how this insect locates food sources, which may aid in future outbreak prediction and better design of grain stores to reduce inwards migration by this insect species.

## MATERIALS AND METHODS

### Insect cultures

*S. zeamais* adults were originally sourced from Malawi and cultured as described in previous studies (*Arnold, Stevenson & Belmain, 2015*; *Jayasekara et al., 2005*) on organic whole wheat grains. The culture was maintained at 25 °C and 60% r.h. in a 14:10 light: dark cycle. Individual adults of known ages and sex were used in experiments, factors

which were included in the data analysis; sex was determined by inspection of the rostrum appearance under a dissecting microscope (*Dobie et al., 1991*). Insects were removed from cultures upon emergence and held in mixed-sex containers, so reproductively mature individuals (over around 4 days old) were assumed to be mated. Unmated and very young adult *S. zeamais* are nonetheless also motivated to forage for food as the adults feed on cereals as well as the larvae (*Ukeh et al., 2012*). Test insects were deprived of food for 0–72 h before use in experiments; the specific length of time was recorded for each individual and both age of insect and duration of food deprivation were included as explanatory variables in the analysis. Each individual was used once only. Experiments took place in a separate room to the main culture, at 26 °C and ambient humidity. Light was provided by high-lux plant growth lamps (irradiance in centre of room: 25.0 $\mu$mol m$^{-2}$ s$^{-1}$; directly beneath camera: 6.5 $\mu$mol m$^{-2}$ s$^{-1}$).

## Servosphere assay

Experiments were carried out using a Syntech TrackSphere LC-300 (Syntech, Hilversum, Netherlands) Servosphere connected to a control unit. A CMOS camera set above the Servosphere provided visual tracking of the insect for locomotion compensation via servomotors. Tracks were recorded in TrackSphere 3.1 (Syntech, Hilversum, Netherlands), which provides both raw and partially-processed data.

The experimental design permitted individual beetles to be tested with one particular visual cue per beetle, under conditions with no odour or blown air; with a clean airstream; and with host odours. A separate cohort of beetles were also tested without any visual stimulus, with otherwise identical no odour/clean air/host odour conditions. Therefore, all condition combinations (with/without visual; with/without odour) were tested.

Odour stimuli were delivered using a vacuum pump that pushed charcoal-filtered air (Agilent Technologies, Wokingham, Berks, UK), through a gas-washing bottle that was either empty or contained 50 g roughly crushed yellow maize (crushed by placing in a plastic bag and crushing with a hammer for 2 min to simulate recently damaged grain (*Arnold, Stevenson & Belmain, 2015*)). Silicone Tygon tubing (Ø internal 0.6 mm) (Tygon, Sigma-Aldrich, St Louis, MO, USA) was used throughout and the flow rate was set at 150 ml/min for all odour sources.

Odours were introduced to the insect at 180° relative to the azimuth of the camera recording display (termed the "upwards" direction by the TrackSphere software) (Fig. 1). Previous research has demonstrated that maize volatiles are attractive to *S. zeamais* (*Arnold, Stevenson & Belmain, 2015*; *Ukeh et al., 2012*), and that the components causing positive chemotaxis include hexanal, (*E*)-2-heptenal, and octanal, particularly when those components are presented as a three-odour blend (*Ukeh et al., 2012*). Furthermore, *Sitophilus* sp. have a preference for maize as a host material, even when individuals were themselves raised on wheat (*Trematerra, Lupi & Athanassiou, 2013*).

The Servosphere was surrounded by a screen of white paper (height 270 mm) on all sides to exclude conflating visual distractions originating from the room. Light was provided by high-lux plant growth lamps (irradiance in centre of room: 25.0 $\mu$mol m$^{-2}$ s$^{-1}$; directly beneath camera: 6.5 $\mu$mol m$^{-2}$ s$^{-1}$); the spectral composition is provided in Fig. S1.

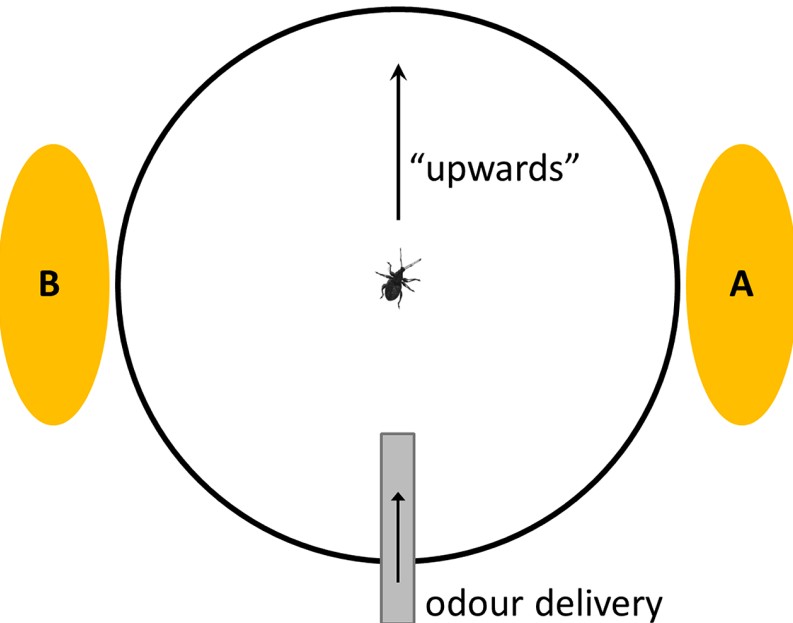

**Figure 1 Schematic of top view of Servosphere setup, showing two alternative positions for visual stimulus (A = 90°; B = 270°) and direction of odour delivery.** Note that odour and visual stimuli are presented perpendicular to each other.

Tested visual stimuli consisted each time of a circle of paper (Ø 153 mm). Use of paper visual stimuli rather than emitting stimuli (e.g. LEDs) was chosen because *S. zeamais* are considered to do the majority of their activities in low light conditions, and therefore emitting stimuli would be ecologically atypical for this species. The paper circles presented one of the following appearances:

1. Coloured circle (one of five colours, with spectral reflectance profiles as in Fig. 2–referred to from here onwards for simplicity as yellow, yellow textured, sand, amber and orange according to their appearance to human eyes). A range of yellow stimuli were tested because yellow has previously been shown to be attractive to *S. zeamais* (*Arnold, Stevenson & Belmain, 2015*) and we sought to test whether this applied to all shades, or only stimuli with particular spectral properties.
2. Black circle–"black."
3. Circle patterned with black and white vertical grating (width of black and white bands equal, 6 mm)–"vertical grating." This explores the concept of high-contrast edges facilitating attraction, and previous studies have shown that vertical black stimuli can elicit attraction behaviours from stored product Coleoptera (*Semeao et al., 2011*).
4. No visual stimulus–"control." This permitted a comparative data set in which the response to odour in the absence of visual stimulus could be tested.

HSV (hue, saturation, value) figures are provided for all the stimuli, and also for wheat and maize, in Table 1, to provide human-relevant context for the appearance of these stimuli. Values for reflectance at 366, 520 and 564 nm are also given for these stimuli, as

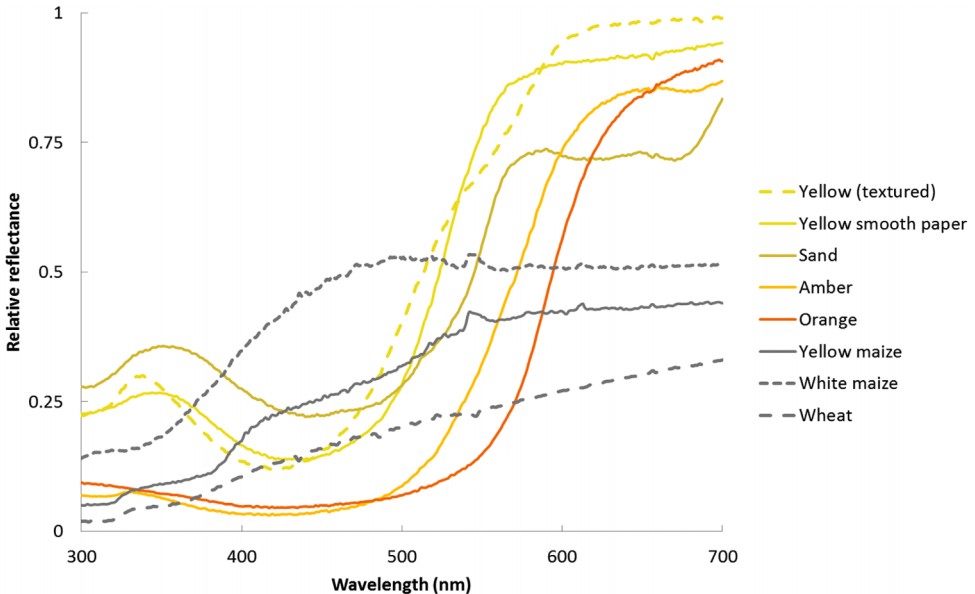

**Figure 2 Spectral reflectance curves for the coloured stimuli presented to the beetles.** As measured on an Avantes AvaSpect-2048 using an Avantes AvaLight-DH-S-BAL relative to a BaSO$_4$ white standard.

**Table 1 Hue, saturation and value figures for coloured paper stimuli and natural host materials, alongside proportional reflectance at 366, 520 and 564 nm.** These being the published peaks of photoreceptor sensitivity in another weevil species, *Rhynchophorus ferrugineus* (*Ilić, Pirih & Belušič, 2016*).

| Stimulus | H | S | V | 366 nm | 520 nm | 564 nm |
|---|---|---|---|---|---|---|
| Yellow | 28 | 92.7 | 74.9 | 0.248018 | 0.461077 | 0.84916 |
| Yellow (textured) | 29 | 94.3 | 75.3 | 0.226178 | 0.547739 | 0.76652 |
| Sand | 22 | 83.3 | 70.6 | 0.349127 | 0.355651 | 0.685641 |
| Amber | 16 | 100 | 100 | 0.050626 | 0.145598 | 0.439424 |
| Orange | 9 | 97.3 | 100 | 0.066094 | 0.090617 | 0.211301 |
| Yellow maize | 30 | 58.9 | 50.6 | 0.099471 | 0.362622 | 0.408071 |
| White maize | 25 | 30.1 | 40.4 | 0.229231 | 0.523592 | 0.505379 |
| Wheat | 24 | 61.3 | 53.7 | 0.059449 | 0.217541 | 0.242793 |

these are the published peaks in spectral sensitivity in the eyes of another pest weevil, *Rhynchophorus ferrugineus* (*Ilić, Pirih & Belušič, 2016*), and are consequently a potential indication of the colour vision a maize weevil may possess. Two spectral receptor types with sensitivities > 500 nm implies the beetles are likely to be able to discriminate green, yellow and orange hues well. While the sand-coloured paper appears superficially most similar to the colours of white and yellow maize, the yellow textured and yellow papers are closer to maize in terms of the hue and value (brightness) measures compared to the other coloured stimuli. The yellow paper also has the highest reflectance at 564 nm out of all the coloured stimuli.

Visual stimuli were presented at 90 or 270° (randomised) to the direction on the Servosphere (Fig. 1). These were positioned at 154 mm from the weevil's location, meaning

that the solid stimuli would subtend a visual angle of 52.4°. Previous research on bees indicates that honeybees and bumblebees can detect the colour of a stimulus subtending 15° (*Dyer, Spaethe & Prack, 2008*; *Giurfa et al., 1996*; *Spaethe, Tautz & Chittka, 2001*), and therefore even with the low-resolution eyes of small insects it can be confidently argued that these stimuli were not only visible to the weevils, but that they should also have been able to detect the colour.

Each insect was placed on the Servosphere and allowed 1 min to acclimate to the new location before commencing motion recording. The insect was then recorded for 15 min in total (sequence below) consisting of five periods of three minutes, in which combinations of visual and/or odour stimuli were presented. The presentation order was: visual stimulus alone, visual stimulus with clean airstream, two periods of visual + odour stimulus, then a final "recovery" period with the visual stimulus alone (it is not practical to alter visual stimuli during the recording period). If the insect flew away (*S. zeamais* can fly but rarely chooses to) the recording was abandoned. The surface of the Servosphere was only handled while wearing gloves and was cleaned regularly using 70% ethanol to prevent chemical residues of previous test animals influencing subsequent animal behaviours.

## Predicted behaviours

Predictions of the angle relative to the camera azimuth that one might expect the beetle to average in its orientation in the case of different visual stimuli are shown in Table 2, based on the principle that the insect will walk towards an attractive stimulus, and when two equally attractive stimuli (of any modality) are presented at different angles, the insect would be expected to choose a path that (on average) is intermediate between them. The odour stimulus used is a known attractant for *S. zeamais* (*Ukeh et al., 2010*; *Ukeh et al., 2012*).

## Statistics

The TrackSphere software provides information about each insect's distance walked (both in total and towards the odour stimulus), direction walked and path straightness. Generalised Linear Models were performed on linear data, using age, sex and period of food deprivation as well as treatment as explanatory variables (Table S1). For comparisons of movement towards the odour source, if a beetle did not move during a particular recording period, a zero distance value was recorded and the beetle was included in the analysis. Period 4 was chosen over period 3 to evaluate responses in the presence of both odour and visual cues, as this assessed the beetle's final choice after having had time to settle on a behavioural response to the dual stimuli and was therefore judged to be more representative of the behavioural preference.

Vectors of movement were calculated from the total X and Y displacement of the insect during each recording period. Statistical analyses of these vectors were performed in SPSS version 20 (IBM, NY, USA), RStudio version 0.97 running R version 3.0.2 (*R Development Core Team, 2008*), and Microsoft Excel for Windows 7, using circular statistics techniques described in *Batschelet (1981)*. Vectors for each beetle during each of the five exposure

**Table 2 Predictions of insect walking vectors.** If visual stimulus is presented at 90° and odour at 180°, and assuming beetle is motivated to seek food. All angles in degrees.

| Nature of visual stimulus | Period 1 (visual alone) | Period 2 (visual + air) | Period 3 (visual + odour 1) | Period 4 (visual + odour 2) | Period 5 |
|---|---|---|---|---|---|
| **Attractive** | 90 | Between 90 and 180 but closer to 90 | Between 90 and 180 | As period 3 | 90 |
| **Neutral** | Random | 180 | 180 | 180 | Random |
| **Repellent** | 270 | 180–270 | 180–270 | 180–270 | 270 |

periods (control 1, airstream, odour 1, odour 2, control 2) were calculated relative to the azimuth values, and "mirror-reversed" in the case of replicates where the insect was presented with the visual cue at 270°, so that all vectors could be directly compared. Mean vectors for each period for each visual stimulus were calculated and tested for significant clustering around the mean via the modified Rayleigh's V-test (*Batschelet, 1981*). This is a standard method of analysis employed in previous Servosphere studies (*Bell & Kramer, 1980*). Using the "circular" package in R, differences between the directional responses to the different colours of stimuli, with and without odour present, were compared using the Watson-Wheeler test. Bonferroni corrections were applied to outputs as appropriate.

To test for directionality, we categorised angular deviation for each insect as "towards the stimulus" or "not towards the stimulus" for periods 1 and 4. "Towards the stimulus" was considered to be any direction between 60 and 120° for the visual stimulus and 150–210° for the odour stimulus when present. For period 4 we also calculated the number of insects displaying an "intermediate direction" of movement, meaning any angular deviation between 90 and 180°. For each visual stimulus type, we then used a binomial test to consider whether the insects were more likely than random (i.e. more than one sixth of the insects for the stimulus/not stimulus or more than one quarter of the insects for the intermediate/not intermediate directionality) to select that direction of movement, with a Bonferroni correction for multiple comparisons.

## RESULTS

In total, 147 individuals were tested, (74 males and 73 females). Thirty-one individuals were tested with the yellow stimulus, 16 yellow textured, 16 sand, 15 amber, 15 orange, and 19 with the black stimulus, 19 with the grating and 17 in the control setup with no visual stimulus. This was expected to provide suitable power to detect differences in orientation angle of 20° between treatments, and differences in distance moved of 13 mm towards odour sources.

In the absence of a discrete coloured visual stimulus, insects oriented towards the odour when it was present (mean angular deviation 155°, r = 0.409, $p = 0.006$) and randomly when it was not (mean angular deviation 137°, r = 0.241, $p = 0.177$). The control confirms that in the absence of a visual stimulus, the beetles do not orient towards the 90° direction on the Servosphere: in the absence of all stimuli the mean angular

**Table 3 Effect of different parameters and their interactions on distance walked by *S. zeamais* during Servosphere recording periods 1, 2 and 4.** Analysed via General Linear Model with listed parameters included as explanatory variables.

| Factor | Has effect? | Colour alone (*p*-value if significant) | Colour + airstream (*p*-value if significant) | Colour + odour (*p*-value if significant) |
|---|---|---|---|---|
| Treatment | Yes | **0.005** | **0.019** | – |
| Sex | No | – | – | – |
| Age | No | – | – | – |
| Time food deprived | Yes | **0.016** | – | – |
| Treatment * Sex | No | – | – | – |
| Treatment * Age | No | – | – | – |
| Age * Sex | No | – | – | – |
| Age * Time food deprived | No | – | – | – |

Note:
Too few degrees of freedom present to return results for: Treatment * Time food deprived; Sex * Time food deprived; Treatment * Sex * Age; Treatment * Sex * Time food deprived; Treatment * Age * Time food deprived; Sex * Age * Time food deprived; Treatment * Sex * Age * Time food deprived.

deviation is random, whereas when a food odour or airstream is present, it is consistently at 180° to the azimuth of the camera display (i.e. towards the odour source).

Table 3 shows the results of a General Linear Model analysis of the whole dataset, with treatment, age, sex and period of food deprivation as independent variables and the distance walked towards the odour source during periods 1–5 as a response variable each time. While responses to odour when colour stimuli were present were not fully consistent, treatment was a significant factor overall in determining odour response (Tables 3 and S1; GLM, Hotelling's Trace, $F_{30,117} = 2.196$, $p = 0.002$). Effects were particularly notable during period 4, when visual and food odours were both present. In period 4, stimuli that appeared attractive in terms of mean vectors also elicited movements towards odour stimuli (with the exception of the vertical grating) (Figs. 3A and 3B), suggesting that combination of odour and colour, even when locations differ, may increase motivation. In addition, the yellow textured stimulus was also associated with movement towards the stimulus. In comparison, amber, orange and sand-coloured stimuli were not associated with movement towards the odour stimulus. There was a significant difference in distance walked towards the odour source in the presence of yellow versus orange stimuli (difference = 252 mm, yellow greater, $p = 0.026$) and yellow versus amber stimuli (difference = 273 mm, yellow greater, $p = 0.013$). Mean distances walked in periods 1 and 4 for the control visual stimulus and yellow, an attractive stimulus, are shown in Fig. 3C.

With respect to mean angular deviations, the directions of insects with and without odour are shown in Fig. 4, with Rayleigh test results presented in Table S2. The black (mean vector 84.4° without odour (Rayleigh test, $z = 15.0$, $p < 0.0001$); 98.5° with odour ($z = 9.49$, $p < 0.0001$)), vertical grating (mean vector 80.1° without odour ($z = 3.42$, $p = 0.0003$); 95.1° with ($z = 3.40$, $p < 0.0001$)) and yellow (mean vector 93.6° without odour ($z = 6.67$, $p < 0.0001$); 145.6° with ($z = 10.2$, $p < 0.0001$)) stimuli all showed significant

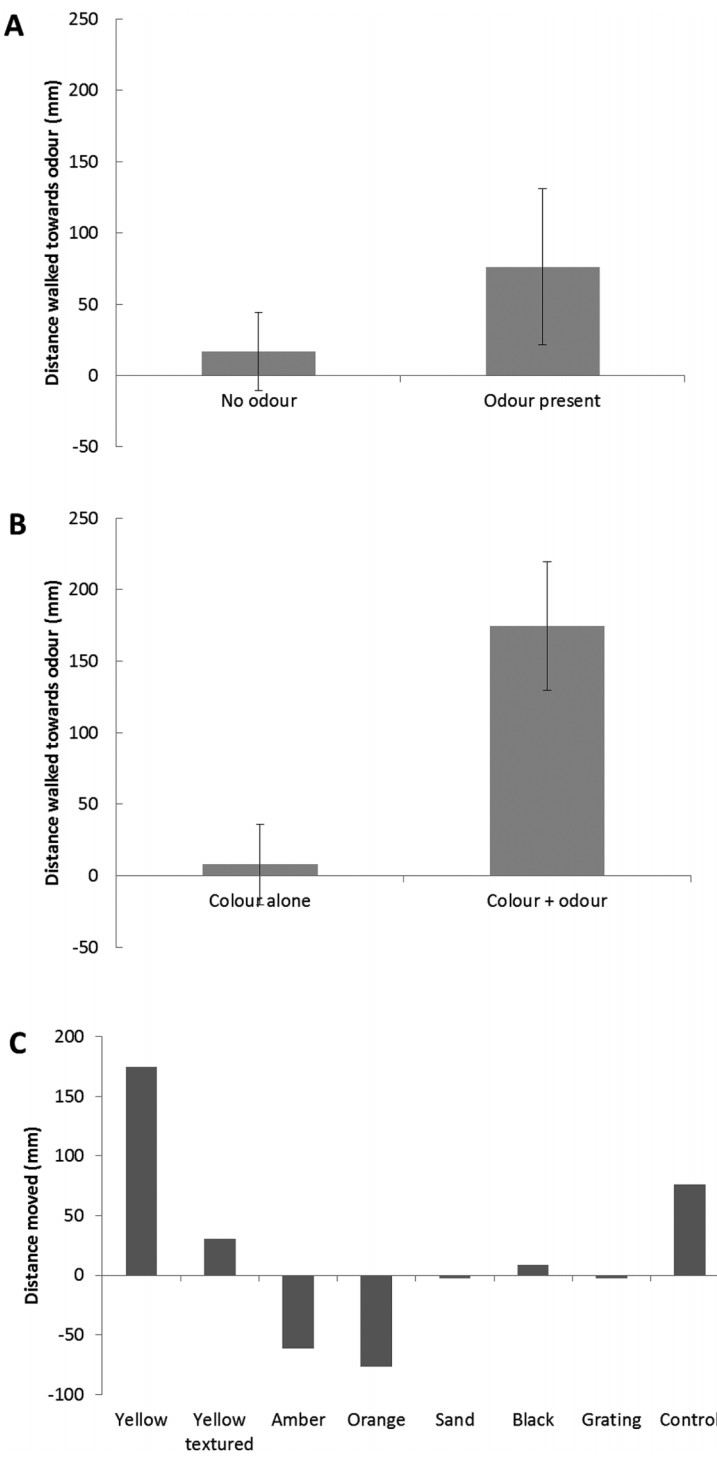

**Figure 3 Distance travelled (mean ± s.e.m.) in direction of odour tube.** (A) by beetles presented with no visual stimulus, with and without odour ($N = 17$ insects) (without = neither blown air nor odour); (B) by beetles presented with the yellow visual stimulus during Period 1 (visual stimulus present, no odour/blown air) and Period 4 (both visual and odour stimuli present) ($N = 31$); (C) distance moved towards odour tube by all beetle cohorts, during Period 4 (both odour and visual stimuli present) ($N = 147$ insects) (negative values indicate net movement away from the tube). ("Control" bar = no visual stimulus presented.).

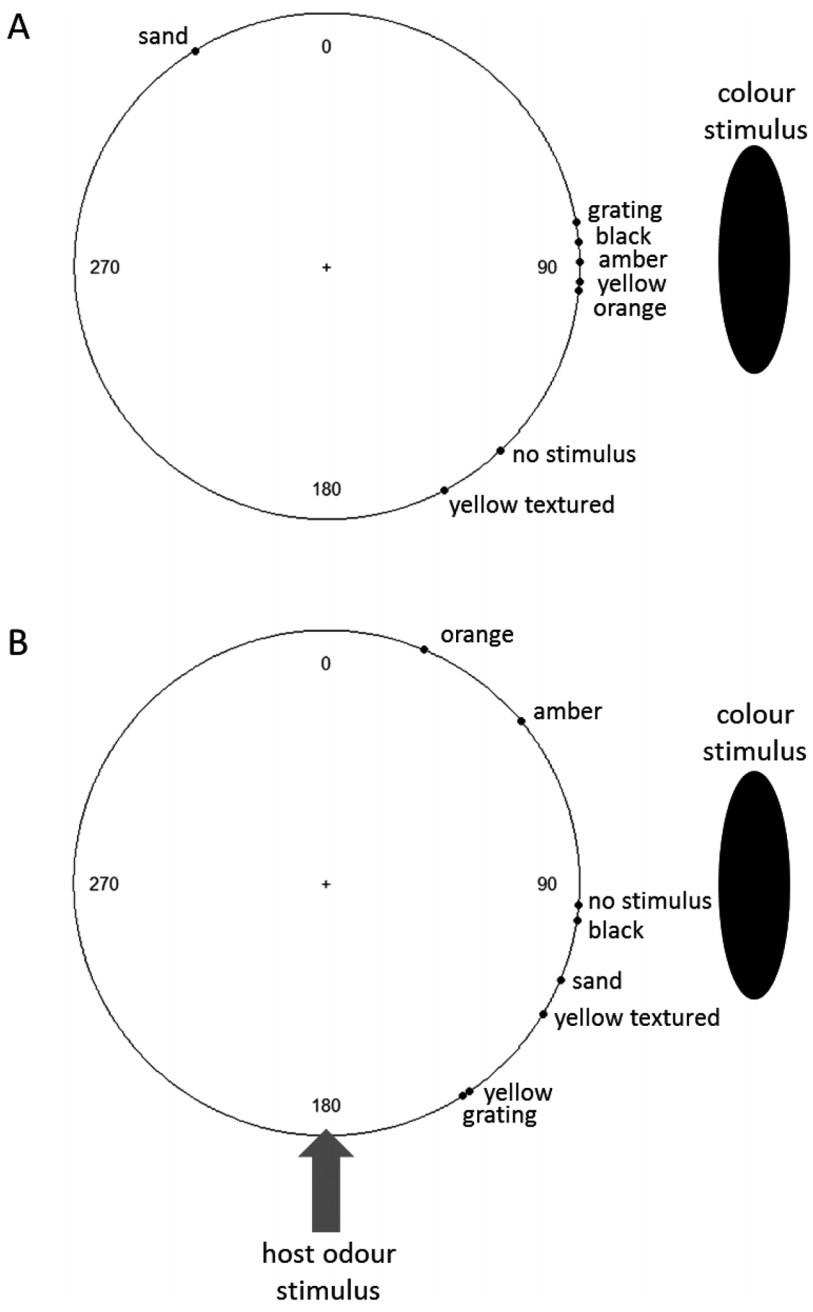

**Figure 4 Mean angles (°) of beetle orientation in the presence of (A) a visual stimulus and no odour or blown air or (B) with both the visual stimulus and an airstream containing host (maize) odour.** "No stimulus" = odour conditions as for other cohorts, but with no visual stimulus presented. ($N = 147$ insects in total).

non-homogeneity (i.e. the insects were not choosing random directions). The binomial analysis of proportion of insects in each case tending to choose a direction towards the stimuli indicated that, in the absence of odour, the black stimulus elicited significant clustering of the angular deviations around 90° (i.e. towards the stimulus source, binomial test, $p < 0.0001$) (clustering towards the yellow stimulus was no longer significant after

Bonferroni correction). When the odour was introduced as well, while insects with the black stimulus present continued to be clustered significantly in their orientation towards it (binomial test, $p < 0.0001$), the insects presented with the yellow stimulus instead were significantly clustered around the odour source (binomial test, $p = 0.0005$) and, in fact, in direction intermediate between the odour and visual stimuli (binomial test, $p = 0.0002$).

The Watson-Wheeler tests revealed that there were differences between the beetles' responses to different colours in terms of the mean vector they chose, both without the presence of any odour cues (Watson-Wheeler test, $W_{12} = 33.31$, $p = 0.00087$) and with both odour and visual cues present, but at right angles to each other ($W_{12} = 27.64$, $p = 0.00624$). This indicates that some colours elicit a stronger behavioural response than others, and in some cases this response is strong enough to override or interfere with the response to odour cues. Pairwise comparisons of colour responses indicate that both with and without the presence of odour, beetles responded significantly differently to black stimuli compared with most other stimuli, and that without an odour cue, the beetles also responded differently to the sand-coloured cue, with the mean vector leading away from the stimulus source; however, as noted above, the vectors are not significantly clustered in the presence of this stimulus, indicating that this was probably random.

Based on the results, black was most consistent with the "attractive" prediction; i.e. the high-contrast, achromatic stimulus produced attraction with and without food odours present. Yellow and vertical grating targets were also attractive stimuli: beetles oriented towards these stimuli in the absence of any confounding odour stimulus. However, when the odour was presented perpendicular to the visual cue, the two stimulus modalities affected beetle behaviour and the path chosen by the beetle was more intermediate between the two sources. This suggests that the odour and visual cues, at this intensity, are of comparable attractiveness. The behaviour of the beetles towards these three visual cues was consistent between individuals, indicated by significant clustering of mean vectors. These data indicate that different shades of yellow are not equally attractive, and high achromatic contrast appears to be equally or more attractive to *S. zeamais* than the chromatic cues presented.

## DISCUSSION AND CONCLUSIONS

Locomotion compensators have been used to study the behaviour of insects in response to attractive odour stimuli (host odours, pheromones, plant volatiles, etc.) (*Becher & Guerin, 2009*; *Otálora-Luna, Perret & Guerin, 2004*). They are of limited use to study repellent odours (we could only find two incidences in the literature (*McMahon, Krober & Guerin, 2003*; *Zermoglio et al., 2015*)) as insects may not always respond to a repellent odour by simply walking in the downstream direction relative to the odour; they may instead attempt to move laterally, stop dead or take flight and abandon the Servosphere. It can therefore be difficult to characterise repellency as a behavioural response on this apparatus. However, as attractive odours will induce a walking insect to orient in an upwind direction, locomotion compensators such as the Servosphere can be used to examine responses to host volatiles and pheromones.

Servospheres have rarely been used to study spectral preferences in visual orientation of walking insects (*Beattie et al., 2011*; *Otálora-Luna & Dickens, 2011*; *Otálora-Luna, Lapointe & Dickens, 2013*; *Van der Ent & Visser, 1991*) and never previously used with spectrally quantified non-emitting stimuli such as coloured paper that might present a controlled but more ecologically relevant motivation. We demonstrate its utility in this context for the first time, presenting evidence that attractive responses to non-emitting visual stimuli can be at least as strong as for odours. Future work could incorporate instantaneous remote control of visual stimuli, permitting increased complexity and evaluation of the effect of adding or removing a visual cue mid-recording. However, the value of using non-emitting stimuli in tests must be highlighted, as coloured lights may elicit unusual behaviour in insects that often forage in dark conditions. While a 360° LED display cylinder around the Servosphere could provide maximal real-time ability to control an insect's visual environment and could permit detailed studies of visually-guided orientation and navigation behaviour in pest species such as *Sitophilus zeamais* (as well as insects such as ants or carabids), the ecological relevance of such a setup must be considered. The dispersal behaviour of *S. zeamais* is not fully characterised, so the timing of it (day versus night) and the visual cues used for navigation by dispersing individuals remain to be discovered. How they respond to point sources of light when dispersing and how often they would be expected to be active when sufficient light is available to make use of colour cues are not known, but this and our previous study (*Arnold, Stevenson & Belmain, 2015*) indicate that the capacity to use colour information in this species is present.

Visual experiments involving the Servosphere have an additional advantage of being able to present multiple visual stimuli simultaneously. It can be used to test relative importance of different stimuli in orientation. Future work could also test additive effects, in a setup where odour and visual stimuli both originate from the same source, though we found that presentation of the visual stimulus can be obstructed by the odour administering tube. The most useful variables for these studies appear to be the mean angular deviation of the insect's movement during each recording period (indicating overall direction of movement), and the upwind distance walked. As the periods are of set duration, this is determined by the mean velocity of the insect's motion in that direction.

Our experiment shows that a coloured stimulus (a yellow circle with some UV-reflectance and a mid-point of the step function around 525 nm) is attractive to *S. zeamais*, but that other shades of yellow are not. Furthermore, monochromatic stimuli (black circle, or black and white grating) are more attractive than the yellow stimulus. This is consistent with many other studies of insects, including pests and animal disease vectors, showing contrast is a key cue for orientation behaviours (*Rockstein, 1974*; *Semeao et al., 2011*). Therefore, it is likely that *S. zeamais* responds both to coloured stimuli and to high-contrast stimuli (black-on-white). This is similar to *Semeao et al. (2011)*'s findings that another pest of stored cereal products, *Tribolium castaneum*, orients towards tall, vertical black shapes. How this behaviour is mediated at the neural level remains unknown. Bees are known to do much of their visual processing using an achromatic channel mediated by the green receptor (e.g., motion, distance vision) (*Giurfa et al., 1996*)

and it is likely that similar mechanisms underpin the vision of other insects, including weevils. In this experiment, the most attractive colour (yellow) had a relatively high ratio of green to blue reflectance, but also moderate UV-reflectance. This suggests that attractiveness of a chromatic stimulus to this species may be influenced by the relative intensities of green, blue and UV reflection of the surface, but further investigation and modelling are required. It is also possible that the intense yellow of this type of paper serves as a supernormal stimulus related to food, eliciting similar positive orientation behaviours to that of yellow in hoverflies (thought perhaps to be a supernormal stimulus response that aids pollen-seeking behaviours) (*Kelber, 2003*) or leafhoppers (thought to aid in seeking foliage) (*Todd, Phelan & Nault, 1990*). Responding strongly to exaggerated stimuli that considerably exceed the intensity of the natural material encountered in nature has been hypothesised as a way to increase success in locating the food substance.

Insect responses to the odour were not entirely consistently attractive. We hypothesised that this may depend on the motivational state of the insect. While period of food deprivation was not a significant factor in determining distance walked towards the odour during period 4 (colour + odour), other factors including interactions with other cues, reproductive state of females or some effect of larval conditions (*Rietdorf & Steidle, 2002*) could also influence the level of motivation a beetle has for orientating towards host material. There also appears to be an interaction with visual cues, as the distance walked towards host odour was typically greater when an attractive visual cue was present–even in a different location to the odour source–than when a less attractive colour was presented. *Otálora-Luna, Lapointe & Dickens (2013)* also found that the visual environment could stimulate higher levels of walking activity, but in that case where the visual and odour cues were presented perpendicularly, the presence of an attractive visual cue overrode the attraction to the odour source. Our result could imply that the presence of an attractive, coloured stimulus may enhance the overall motivation of beetles to seek food.

This study underlines the importance of visual cues in host location by pest insects, both in terms of contrast and chromaticity–even in insects normally associated with poorly lit environments. The interaction between odour and colour in orientation towards targets is very important in various insect taxa (*Raguso & Willis, 2005*; *Wäckers & Lewis, 1994*); optimising both types of cue can be used to enhance the efficacy of trapping and monitoring devices. Visual and odour cues can be synergistic when presented together; they can also operate at different distances from the source, with insects responding to visual cues further away and odour at closer range (*Frye, Tarsitano & Dickinson, 2003*), or vice versa. It is evident that while some visual appearances may enhance the effectiveness of traps or, conversely, deterrents, other colours or patterns will be less effective. The most attractive colours may not necessarily correspond perfectly to the colours of host material.

## ACKNOWLEDGEMENTS

We thank Natalie Morley for technical support with insect cultures and Stephen Young for statistical advice. We thank James Campbell and an anonymous referee for their thoughtful comments.

## Funding

The authors received funding from a University of Greenwich Higher Education Funding Council for England grant to SEJA, the McKnight Foundation supported project "Safe and effective pesticidal plants for agro-ecological intensification of legumes", and the European Union African-Caribbean-Pacific Science and Technology Programme OPTIONs project FED/2013/329-272. The funders had no role in study design, data collection and analysis, decision to publish, or preparation of the manuscript.

## Grant Disclosures

The following grant information was disclosed by the authors:
University of Greenwich Higher Education Funding Council for England Grant to SEJA.
McKnight Foundation supported project "Safe and Effective Pesticidal Plants for Agro-Ecological Intensification of Legumes".
European Union African-Caribbean-Pacific Science and Technology Programme OPTIONs Project: FED/2013/329-272.

## Competing Interests

The authors declare that they have no competing interests.

## Author Contributions

- Sarah E.J. Arnold conceived and designed the experiments, performed the experiments, analyzed the data, wrote the paper, prepared figures and/or tables, reviewed drafts of the paper.
- Philip C. Stevenson contributed reagents/materials/analysis tools, wrote the paper, reviewed drafts of the paper, involved in interpretation of results.
- Steven R. Belmain contributed reagents/materials/analysis tools, wrote the paper, reviewed drafts of the paper, involved in interpretation of results.

## Data Deposition

   No raw genetic/molecular/bioinformatics data were generated by the research.
Behavioural data raw numbers are supplied as Supplementary Information.

## Supplemental Information

Supplemental information for this article can be found online at http://dx.doi.org/10.7717/peerj.2219#supplemental-information.

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
