# Peer review of "Shades of yellow: interactive effects of visual and odour cues in a pest beetle"

_PeerJ, doi:10.7717/peerj.2219_

## Round 0.1 · original submission · Major Revisions

This is an interesting study, but with divergent views from both reviewers, I am giving you the opportunity to review the manuscript. Please make sure you fully address all the queries and comments made by both reviewers.

Reviewer 1 ·

Basic reporting

This manuscript reports a series of experiments testing bimodal orientation in a weevil pest of maize. Even thought the subject is interesting and deserves to be investigated, the manuscript has series of flaws both, conceptual and methodological, that weakens the effort made by the authors. Besides, the goal of the work is difficult to assess. If it is the utility of servosphere for studying insect orientation (i.e. the reason why they were developed and commercialised since decades), authors should have gone deeper in their advantages and limitations and, even more important, the theory behind open-loop experiments. If the goal was testing colour vision, authors should have employed appropriate conditions (e.g. they describe the reflectance of stimuli, but not the characteristics of the incident light during the experiments). If the aim of the study was testing multimodal orientation, then authors should have discussed theoretical concepts and protocols for quantitative analysis of stimuli interactions, as has been done by Horn and others decades ago.
On the other hand, the use of an imprecise language (e.g. “upwards”, when insects walk on a horizontal plane, “behavioural inclinations”), gives the impression to this reviewer of a quite inexperienced and superficial approach to the problem.

Experimental design

My first impression about the experimental design is that authors have not considered at all the theory behind the development and employ of locomotion compensators. They present a series of practical arguments or how they were used by other authors, but nothing is said about what means conducting an experiment in open- vs. closed-loop conditions, and the kind of information that each approach may provide to the experimenter.
Repeatedly, authors refer to colour vision, but they never define it. This is very important, because one would immediately see that the experimental conditions do not allow testing or concluding about colour vision. For instance, authors present the human characterization of the colours used (hue, saturation and value), and the reflectance of the papers (even in the UV). Nevertheless, they never describe the light source in terms of intensity and spectral quality; which is basic for reflecting visual stimuli. This reveals (once more) the superficial approach and intuitive argumentation that characterize the whole manuscript.
There is a number of instances through the manuscript where the reporting of the statistical analysis is not fully satisfying. This concerns both, the methodological description and, more notably, the reporting of results. Especially when models are as complex as in some of the analyses performed here, it is essential that all variables, factors, interactions, etc. are clearly identified and their use justified. In the same line, I would have been glad to see tables reporting the full model outputs (including degrees of freedom etc.).
This concerns also the use of circular statistics. The Rayleigh-test only evaluates whether or not data follow a uniform distribution over 360°, but it does not test for a preferred direction, as indicated in lines 269-277. It is worth noting that in circular statistics “non-uniform” does not mean “oriented in a particular direction”. Watson-Wheeler test for homogeneity on two or more samples of circular data; the difference between the samples can be in either the mean or the variance. Other relevant details are missing (e.g., how vectors were calculated, how data coming of the same individual were considered, etc.), necessary to exclude pseudoreplication, which is a very common error in this kind of studies.
I will stop here, but there are many others.

Validity of the findings

The numerous conceptual and methodological problems make very difficult to asses the validity of findings. Crucial information is missing about the stimuli tested, the experimental procedures and the analysis of data.

Additional comments

It is clear that authors are interested in applied entomology and they try to conclude about practical aspects, rather than on mechanisms. This is not a problem. However, when biological problems as colour vision, multisensory integration or orientation mechanisms are analysed, it is essential a minimal conceptual background and a basic comprehension of methodological approaches. I encourage the authors to refer to the abondant literature on these subjects.

·

Basic reporting

No Comments

Experimental design

No Comments

Validity of the findings

No Comments

Additional comments

I found this to be a very interesting study that takes a novel approach to evaluate the interactions between visual and odor cues. The experimental design and analysis was well laid out and the manuscript was clearly written. I have no major concerns or revisions. A few things to consider are listed below.

1. The spectral reflectance curves are shown for the colored papers and maize and wheat (potential food items), which is very helpful to see. But it would be interesting to have some more discussion of how the colors the weevils respond to behaviorally in your bioassay relate to potential for response to food items. The curves don't seem to be that similar between maize and wheat and the color papers.
2. In Figure 4 C, is this the total distance moved or as in A and B is it distance toward the odor? Could make this clearer. The negative distances for the amber and orange color and low values for every treatment but yellow compared to the control are interesting. This is an interesting effect - as you mention in the discussion - since it suggest that the general color environment may impact response to food odor, not just an additive impact of combing a color and odor together.
3. Not sure Figure 3 is needed, since this is already in the text.
4. Mention that Sitophilus tend to be found primarily under low light conditions as rationale for using colored paper versus lights. I think the use of paper is a good approach, but suggest that given the dispersal ability of this species in the outside environment and tendency to exploit small patches of grain they may be in bright light conditions more regularly than expected and response to color may be related to responses under these conditions as well, not just low light.

Overall a very interesting study that leads to some new insights and some new areas of investigation.

---

## Round 0.2 · Minor Revisions

Both reviewers and i are happy with the changes you have made to your manuscript - i congratulate you on your efforts. Please consider the minor corrections that have been identified by Reviewer 2 and resubmit for final approval

Reviewer 1 ·

Basic reporting

Authors have adequately acknowledged my comments and improved the manuscript accordingly.

Experimental design

The experimental design is adequate and this new version has much improved its rationale, underlining the utility of locomotion compensators.

Validity of the findings

Experiments have been well conducted and the results obtained are solid.

Additional comments

This new version has much improved the presentation of your work.

·

Basic reporting

no comments

Experimental design

no comments

Validity of the findings

no comments

Additional comments

I think the authors have done a good job in addressing the concerns raised in the previous round of reviews. While there is always more that can be done, I think it is a stronger paper that is suitable for publication. Don't have any major concerns, but below are a few minor corrections and a comment.
Line 196. “…of odour cues actually…” instead of “…odour cues acted increased…”
Line 203. Change to ‘preferentially on a’
Line 212. Think the ‘and’ after ‘odour cues’ should be deleted.
Line 273. Might reword to ‘…light was provided by high-lux plant growth lamps…, reads a little awkwardly
Line 319. Extra comma and spaces on this line
Line 522-525. The strong response to black circle, is also similar to the response of another stored grain pest the red flour beetle, Tribolium castaneum, that Semeao et al. (2011) Journal of Stored Products Research 47:88-94 found had a strong response to tall black vertical shapes.

---

## Round 0.3 · accepted · Accept

I am happy with the changes you have made, so now it is good to go!